# FBSDiff: Plug-and-Play Frequency Band Substitution of Diffusion Features for Highly Controllable Text-Driven Image Translation

## ABSTRACT

Large-scale text-to-image diffusion models have been a revolutionary milestone in the evolution of generative AI and multimodal technology, allowing extraordinary image generation with natural-language text prompts. However, the issue of lacking controllability of such models restricts their practical applicability for real-life content creation, for which attention has been focused on leveraging a reference image to control text-to-image synthesis. This paper contributes a concise and efficient approach that adapts the pre-trained text-to-image (T2I) diffusion model to the image-to-image (I2I) paradigm in a plug-and-play manner, realizing high-quality and versatile text-driven I2I translation without any model training, model fine-tuning, or online optimization. To guide T2I generation with a reference image, we propose to model diverse guiding factors with different frequency bands of diffusion features in DCT spectral space, and accordingly devise a novel frequency band substitution layer that dynamically substitutes a certain DCT frequency band of diffusion features with the corresponding counterpart of the reference image along the reverse sampling process. We demonstrate that our method flexibly enables highly controllable text-driven I2I translation both in the guiding factor and guiding intensity of the reference image, simply by tuning the type and bandwidth of the substituted frequency band, respectively. Extensive experiments verify the superiority of our approach over related methods in image translation visual quality and versatility.

## CCS CONCEPTS

• **Computing methodologies** → **Image processing**; *Image representations*; Computational photography.

## KEYWORDS

Image-to-image translation, Image manipulation, Diffusion model

## 1 INTRODUCTION

Text-driven I2I translation is an appealing computer vision problem that aims to translate a reference image with open-domain text prompts, and is also a typical application of the booming multimodal technology. Since the advent of CLIP [29] bridging vision and language with large-scale contrastive pre-training, attempts have been made to instruct image manipulation with text by combining CLIP with generative models. VQGAN-CLIP [6] pioneers

*ACM MM, 2024, Melbourne, Australia*
© 2024 Copyright held by the owner/author(s). Publication rights licensed to ACM.
ACM ISBN 978-x-xxxx-xxxx-x/YY/MM
https://doi.org/10.1145/nnnnnnn.nnnnnnn

**Unpublished working draft. Not for distribution.**

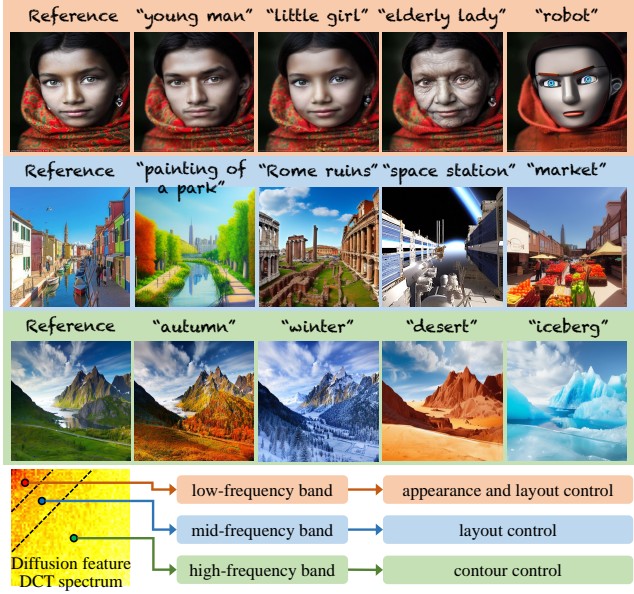

**Figure 1: Based on the pre-trained text-to-image diffusion model, FBSDiff enables efficient text-driven image-to-image translation by proposing a plug-and-play reference image guidance mechanism, which allows flexible control over different guiding factors (e.g., appearance, layout, contour) of the reference image to the generated image by dynamically substituting different types of DCT frequency bands during the sampling process. Better viewed with zoom-in.**

text-driven image translation by optimizing VQGAN [9] image embedding with CLIP image-text similarity loss. DiffusionCLIP [16] fine-tunes diffusion model [12] under CLIP loss to manipulate an image with a text. DiffuseIT [17] combines VIT-based structure loss [39] and CLIP-based semantic loss to guide the diffusion sampling process via manifold constrained gradient [5], synthesizing translated image that complies with the target text while maintaining the structure of the reference image. However, these methods are not competitive in generation quality due to limited model capacity and training data of the backbone generative model.

To promote image translation visual quality, efforts have been made to train large models on massive data. InstructPix2Pix [2] employs GPT-3 [3] and Stable Diffusion [31] to synthesize huge amounts of paired training data, based on which trains a supervised text-driven I2I mapping for general image manipulation tasks. Design Booster [37] trains a latent diffusion model [31] conditioned on a joint representation that fuses both text embedding and image embedding, realizing layout-preserved text-driven I2I translation. Nevertheless, these methods are remarkably computationally intensive due to the need for training large models on immense data.

To circumvent formidable training costs, research has been focused on leveraging off-the-shelf large-scale T2I diffusion models for text-driven I2I translation. This type of methods further divide into fine-tuning-based methods and inversion-based methods.

The former type of fine-tuning-based methods represented by SINE [45] and Imagic [15] fine-tune the pre-trained T2I diffusion model to reconstruct the reference image before manipulating it with a target text. These methods require separate fine-tuning of the entire model for each time of image editing, which is less efficient and prone to underfitting or overfitting to the reference image.

The latter type of inversion-based methods invert the reference image into diffusion model Gaussian noise space and then generate the translated image via diffusion sampling process guided by the target text. A pivotal challenge of this pipeline is that the sampling trajectory may severely deviate from the inversion trajectory due to the error accumulation caused by the classifier-free guidance technique [13], which impairs the correlation between the reference and the translated image. To remedy this issue, Null-text Inversion [23] optimizes unconditional null-text embedding to align the sampling trajectory to the inversion trajectory. Prompt Tuning Inversion [8] and StyleDiffusion [18] minimize trajectory divergence by learning to encode the information of the reference image into learnable conditional embedding. Pix2Pix-zero [26] penalizes trajectory deviation by matching cross-attention maps between the two trajectories with least-square loss. These methods apply online optimization at each diffusion time step to calibrate the whole sampling trajectory, introducing additional time overhead. Moreover, most of these methods rely on cross-attention control introduced by P2P [11] for structure preservation, requiring paired source text of the reference image, which is not available in most cases. PAP [40] maintains image structure by extracting and injecting the internal feature maps and self-attention maps of the denoising U-Net into the reverse sampling trajectory, realizing optimization-free text-driven I2I translation, though the designed feature extraction and manipulation pipeline is heuristic, cumbersome, and time-consuming.

In this paper, we propose a concise and efficient approach termed FBSDiff, realizing plug-and-play and highly controllable text-driven I2I translation from a frequency-domain perspective. To guide T2I generation with a reference image, a key missing ingredient of existing methods is the mechanism to control the guiding factor (e.g., image appearance, layout, contour) and guiding intensity. Since the guiding factors of the reference image are difficult to isolate in the spatial domain but are decomposable in the frequency domain, we consider modeling different guiding factors with the corresponding frequency bands of diffusion features in the Discrete Cosine Transform (DCT) spectral space. Based on this motivation, we propose an inversion-based text-driven I2I framework characterized by a novel frequency band substitution mechanism, which realizes plug-and-play and controllable reference image guidance by dynamically substituting a certain DCT frequency band of diffusion features with the corresponding counterpart of the reference image along the reverse sampling process. As displayed in Fig. 1, T2I image synthesis with appearance and layout control, pure layout control, and contour control of the reference image to the generated image can be realized by substituting the low-frequency band, mid-frequency band, and high-frequency band, respectively, and thus allowing highly controllable and versatile text-driven I2I translation.

The strengths of our approach are fourfold: (I) dynamic reference image control at inference time, realizing plug-and-play text-driven I2I translation; (II) conciseness and efficiency, our method dispenses with the need for paired source text as well as cumbersome attention modulations as compared with existing methods, while still achieving leading I2I performance; (III) more generic methodology, our method applies frequency band transplantation on the denoised features along the reverse sampling trajectory, requiring no access to any internal feature embedding of the denoising network, and thus decouples with the specific diffusion model backbone as contrasted with existing methods; (IV) our method allows to flexibly control the guiding factor and guiding intensity of the reference image simply by tuning the type and bandwidth of the substituted frequency band. The effectiveness of our method is fully demonstrated with both qualitative and quantitative evaluations. To summarize, we make the following key contributions:

- We provide new insights about controllable diffusion process from a novel frequency-domain perspective.
- We propose a novel frequency band substitution technology, realizing plug-and-play text-driven I2I translation without any model training, fine-tuning, or online optimization.
- We contribute a concise and efficient text-driven I2I framework that is free from source text and cumbersome attention modulations, highly controllable in both guiding factor and guiding intensity of the reference image, and invariant to the used diffusion model backbone, all while achieving superior I2I translation performance among existing methods.

## 2 RELATED WORK

### 2.1 Diffusion Model

Since the advent of DDPM [12], diffusion model has soon dominated the family of generative models [7]. DDIM [36] and its variants [20, 20] substantially accelerate diffusion model sampling process. Palette [32] extends diffusion model to the realm of conditional image synthesis. Large-scale T2I diffusion models [25, 30, 33] bring image creation to an unprecedented level, whose computation overhead is significantly reduced by LDM [31] by training diffusion model in low-dimensional feature space. ControlNet [43] and T2i-adapter [24] add spatial control to T2I diffusion models by training a control module conditioned on certain image priors. SDXL [28] and DiTs [27] improve diffusion model backbone to larger capacity. Now, diffusion model has been applied to a wide variety of vision tasks with noticeable performance gains [1, 19, 21, 22, 34, 38, 42], and is still making rapid progress in theory and application.

### 2.2 Computer Vision in Frequency Perspective

Neural networks are mostly used to tackle vision tasks in the spatial or temporal domain, some research improves model performance from a frequency-domain perspective. For example, Ghosh et al. [10] introduce DCT to CNN to accelerate network convergence. Xie et al. [41] propose a frequency-aware dynamic network for lightweight image super-resolution. Cai et al. [4] impose Fourier frequency spectrum consistency to image translation tasks for better identity preservation. FreeU [35] improves image generation quality by selectively enhancing or depressing different frequency components of diffusion model U-Net features.

Figure 2: Overview of FBSDiff. The whole framework contains an inversion process that inverts the reference image into the Gaussian noise space of the latent diffusion model, based on which a reconstruction process is applied to reconstruct the reference image, providing intermediate denoised results as pivotal guidance features that guide the text-driven sampling process by dynamically transplanting certain DCT frequency bands with frequency band substitution layer.

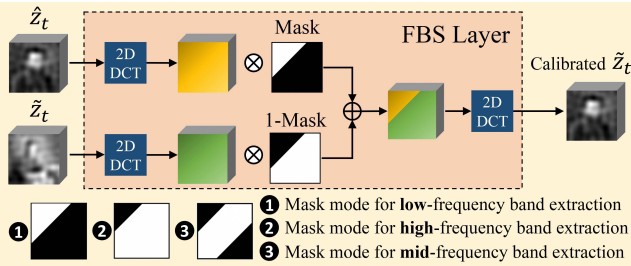

Figure 3: Illustration of the proposed frequency band substitution (FBS) layer. The FBS layer takes in two diffusion features and substitutes a certain DCT frequency band of one feature with the corresponding frequency band of the other feature, where the frequency band extraction and transplantation are implemented with binary masking.

## 3 METHOD

In this section, we first describe the overall model architecture, then elaborate on the frequency band substitution mechanism, and finally summarize the algorithm and show implementation details. For the diffusion model background, please refer to the appendix.

### 3.1 Overall Architecture

Built on the pre-trained Latent Diffusion Model (LDM), FBSDiff adapts it from T2I generation to text-driven I2I translation by proposing a plug-and-play reference image guidance mechanism, realizing controllable guiding factor and guiding intensity of the reference image via dynamic frequency band substitution.

As Fig. 2 shows, FBSDiff comprises three diffusion trajectories: (i) inversion trajectory ($z_0 \rightarrow z_{T_{inv}}$); (ii) reconstruction trajectory ($z_{T_{inv}} = \hat{z}_T \rightarrow \hat{z}_0 \approx z_0$); (iii) sampling trajectory ($\tilde{z}_T \rightarrow \tilde{z}_0$). Starting from the initial feature $z_0 = E(x)$ extracted from the reference image $x$ by the encoder $E$, a $T_{inv}$-step DDIM inversion is employed to project $z_0$ into the Gaussian noise latent space conditioned on the null-text embedding $v_\emptyset$, based on the assumption that the ODE

process can be reversed in the limit of small steps:

$$z_{t+1} = \sqrt{\bar{\alpha}_{t+1}} f_\theta(z_t, t, v_\emptyset) + \sqrt{1 - \bar{\alpha}_{t+1}} \epsilon_\theta(z_t, t, v_\emptyset), \quad (1)$$

$$f_\theta(z_t, t, v_\emptyset) = \frac{z_t - \sqrt{1 - \bar{\alpha}_t} \epsilon_\theta(z_t, t, v_\emptyset)}{\sqrt{\bar{\alpha}_t}}, \quad (2)$$

where $\{\bar{\alpha}_t\}$ are schedule parameters that follows the same setting as DDPM [12], $\epsilon_\theta$ is the denoising U-Net of the pre-trained LDM. The Gaussian noise $z_{T_{inv}}$ obtained after the $T_{inv}$-step DDIM inversion is directly used as the initial noise feature of the subsequent reconstruction trajectory, which is a $T$-step DDIM sampling process that reconstructs $\hat{z}_0 \approx z_0$ from the inverted noise feature $\hat{z}_T = z_{T_{inv}}$:

$$\hat{z}_{t-1} = \sqrt{\bar{\alpha}_{t-1}} f_\theta(\hat{z}_t, t, v_\emptyset) + \sqrt{1 - \bar{\alpha}_{t-1}} \epsilon_\theta(\hat{z}_t, t, v_\emptyset), \quad (3)$$

in which $f_\theta(\hat{z}_t, t, v_\emptyset)$ follows the same form as Eq. 5. The length of the reconstruction trajectory could be much smaller than that of the inversion trajectory (i.e., $T \ll T_{inv}$) to save inference time. The same null-text embedding $v_\emptyset$ is conditioned in the reconstruction trajectory to ensure feature reconstructability (i.e., $\hat{z}_0 \approx z_0$).

Meanwhile, an equal-length sampling trajectory is applied in parallel with the reconstruction trajectory to synthesize the target image. The sampling trajectory is also a $T$-step DDIM sampling that progressively denoises a randomly initialized Gaussian noise feature $\tilde{x}_T \sim \mathcal{N}(0, I)$ into $\tilde{x}_0$ conditioned on the target-text embedding $v$. To amplify the effect of text guidance, classifier-free guidance technique [13] is utilized which interpolates conditional (target-text) and unconditional (null-text) noise prediction at each time step with a guidance scale $\omega$ during the sampling trajectory:

$$\tilde{z}_{t-1} = \sqrt{\bar{\alpha}_{t-1}} f_\theta(\tilde{z}_t, t, v, v_\emptyset) + \sqrt{1 - \bar{\alpha}_{t-1}} \epsilon_\theta(\tilde{z}_t, t, v, v_\emptyset), \quad (4)$$

$$f_\theta(\tilde{z}_t, t, v, v_\emptyset) = \frac{\tilde{z}_t - \sqrt{1 - \bar{\alpha}_t} \epsilon_\theta(\tilde{z}_t, t, v, v_\emptyset)}{\sqrt{\bar{\alpha}_t}}, \quad (5)$$

$$\epsilon_\theta(\tilde{z}_t, t, v, v_\emptyset) = \omega \cdot \epsilon_\theta(\tilde{z}_t, t, v) + (1 - \omega) \cdot \epsilon_\theta(\tilde{z}_t, t, v_\emptyset). \quad (6)$$

Due to the inherent property of DDIM inversion and DDIM sampling, the reconstruction trajectory forms a deterministic denoising mapping towards the reference image. Therefore, the intermediate denoising results $\{\hat{z}_t\}$ along the reconstruction trajectory can be used as pivotal guidance features to calibrate the corresponding

counterpart $\{\tilde{z}_t\}$ along the sampling trajectory to establish the correlation between the reference image and the generated image, and thus enables text-driven I2I translation. Specifically, we implement feature calibration by inserting a plug-and-play frequency band substitution (FBS) layer in between the reconstruction trajectory and the sampling trajectory. FBS layer substitutes a certain frequency band of $\tilde{z}_t$ with the same frequency band of $\hat{z}_t$ along the sampling process to impose a certain guiding effect of the reference image, where both the guiding factor (e.g., appearance, layout, contour) and guiding intensity are flexibly controllable by tuning the type and bandwidth of the substituted frequency band, respectively.

To balance image guidance and generation quality, we partition the sampling trajectory into a calibration phase and a non-calibration phase separated by the time step $\lambda T$. In the former calibration phase ($\tilde{z}_T \rightarrow \tilde{z}_{\lambda T}$), dynamic frequency band substitution is applied at each time step for stable calibration of the sampling process; in the latter non-calibration phase ($\tilde{z}_{\lambda T-1} \rightarrow \tilde{z}_0$), we remove FBS layer to avoid over-constrained sampling results, fully unleashing the generative power of the diffusion model to improve image generation quality. Here $\lambda$ denotes the ratio of the length of the non-calibration phase to that of the entire sampling trajectory.

At last, the final result $\tilde{z}_0$ of the sampling trajectory is converted to the translated image $\tilde{x}$ via the decoder $D$, i.e., $\tilde{x} = D(\tilde{z}_0)$.

## 3.2 Frequency Band Substitution Layer

As Fig. 3 illustrates, the FBS layer takes in a pair of diffusion features $\hat{z}_t$ and $\tilde{z}_t$, converts them from the spatial domain into the frequency domain with 2D-DCT, then transplants a certain frequency band in the DCT spectrum of $\hat{z}_t$ to the same position in the DCT spectrum of $\tilde{z}_t$. Finally, 2D-IDCT is applied to transform the fused DCT spectrum back into the spatial domain as the calibrated $\tilde{z}_t$.

In 2D DCT spectrum, elements with smaller coordinates (nearer to the top-left origin) encode lower-frequency image information, larger-coordinate elements correspond to higher-frequency image components, and most of the DCT spectral energy is occupied by a small proportion of low-frequency elements.

In the FBS layer, the sum of 2D coordinates is used as thresholds to extract DCT frequency bands of different types and bandwidths through binary masking. Specifically, we design three types of binary masks which are respectively termed the low-pass mask ($Mask_{lp}$), high-pass mask ($Mask_{hp}$), and mid-pass mask ($Mask_{mp}$):

$$\begin{cases} Mask_{lp}(x, y) = 1 \ if \ x + y \leq th_{lp} \ else \ 0, \\ Mask_{hp}(x, y) = 1 \ if \ x + y > th_{hp} \ else \ 0, \\ Mask_{mp}(x, y) = 1 \ if \ th_{mp1} < x + y \leq th_{mp2} \ else \ 0, \end{cases}$$

where $th_{lp}$ is the threshold of the low-pass filtering; $th_{hp}$ is the threshold of the high-pass filtering; $th_{mp1}$ and $th_{mp2}$ are respectively the lower and upper bound of the mid-pass filtering. Given a binary mask $Mask_* \in \{Mask_{lp}, Mask_{hp}, Mask_{mp}\}$, the frequency band substitution operation in the FBS layer can be formulated as:

$$\tilde{z}_t = IDCT(DCT(\hat{z}_t) \cdot Mask_* + DCT(\tilde{z}_t) \cdot (1 - Mask_*)), \quad (7)$$

where $DCT$ and $IDCT$ refers to the 2D-DCT and 2D-IDCT transformations respectively, which are described in detail in the **Supplementary Materials**. The usage of the low-pass mask $Mask_{lp}$, high-pass mask $Mask_{hp}$, and mid-pass mask $Mask_{mp}$ respectively corresponds to the extraction and substitution of the low-frequency

---

**Algorithm 1** Complete algorithm of FBSDiff

---

**Input:** the reference image $x$ and the target text.
**Output:** the translated image $\tilde{x}$.
1: Extract the initial latent feature $z_0 = E(x)$.
2: **for** $t = 0$ to $T_{inv} - 1$ **do**
3:    compute $z_{t+1}$ from $z_t$ via Eq. 1;
4: **end for**{DDIM inversion}
5: Initialize $\tilde{z}_T$ from the isotropic Gaussian distribution.
6: **for** $t = T$ to $\lambda T + 1$ **do**
7:    compute $\hat{z}_{t-1}$ from $\hat{z}$ via Eq. 3;
8:    compute $\tilde{z}_{t-1}$ from $\tilde{z}$ via Eq. 4;
9:    calibrate $\tilde{z}_{t-1}$ with $\hat{z}_{t-1}$ via Eq. 7;
10: **end for**{DDIM sampling in the calibration phase}
11: **for** $t = \lambda T$ to 1 **do**
12:    compute $\tilde{z}_{t-1}$ from $\tilde{z}$ via Eq. 4;
13: **end for**{DDIM sampling in the non-calibration phase}
14: Obtain $\tilde{z}_0$ and the final translated image $\tilde{x} = D(\tilde{z}_0)$.

---

band, high-frequency band, and mid-frequency band, which controls different guiding factors of the reference image to the finally generated image:

- **Low-frequency band** substitution enables low-frequency information guidance of $x$, realizing appearance (e.g., color, luminance) and layout control over the generated image $\tilde{x}$.
- **High-frequency band** substitution enables high-frequency information guidance of the reference image $x$, realizing contour control over the generated image $\tilde{x}$.
- **Mid-frequency band** substitution enables mid-frequency information guidance of the reference image $x$. By filtering out higher-frequency contour information and lower-frequency appearance information in the DCT spectrum, it realizes pure layout control over the generated image $\tilde{x}$.

The DCT masking type and the corresponding thresholds used in the FBS layer are algorithm hyper-parameters, which could be flexibly modulated to enable diverse guiding factors and continuous guiding intensity of the reference image to the generated image.

## 3.3 Implementation Details

We use the pre-trained Stable Diffusion v1.5 as backbone diffusion model and set the classifier-free guidance scale $\omega = 7.5$. We use 1000-step DDIM inversion to ensure high-quality reconstruction, i.e., $T_{inv}$=1000, and use 50-step DDIM sampling for the reconstruction and sampling trajectory, i.e., $T$=50. Along the sampling trajectory, we allocate 55% time steps to the calibration phase and the remaining 45% steps for the non-calibration phase, i.e., $\lambda$=0.45. For the default DCT masking thresholds used in the FBS layer, we set $th_{lp}$=80 for the low-frequency band substitution (low-FBS); $th_{hp}$=5 for the high-frequency band substitution (high-FBS); $th_{mp1}$=5, $th_{mp2}$=80 for the mid-frequency band substitution (mid-FBS). The complete algorithm of FBSDiff is described in Alg. 1.

## 4 EXPERIMENTS

In this section, we first present and analyze the qualitative results of our method; then delve into the frequency band substitution with ablation studies; and finally show quantitative evaluations.

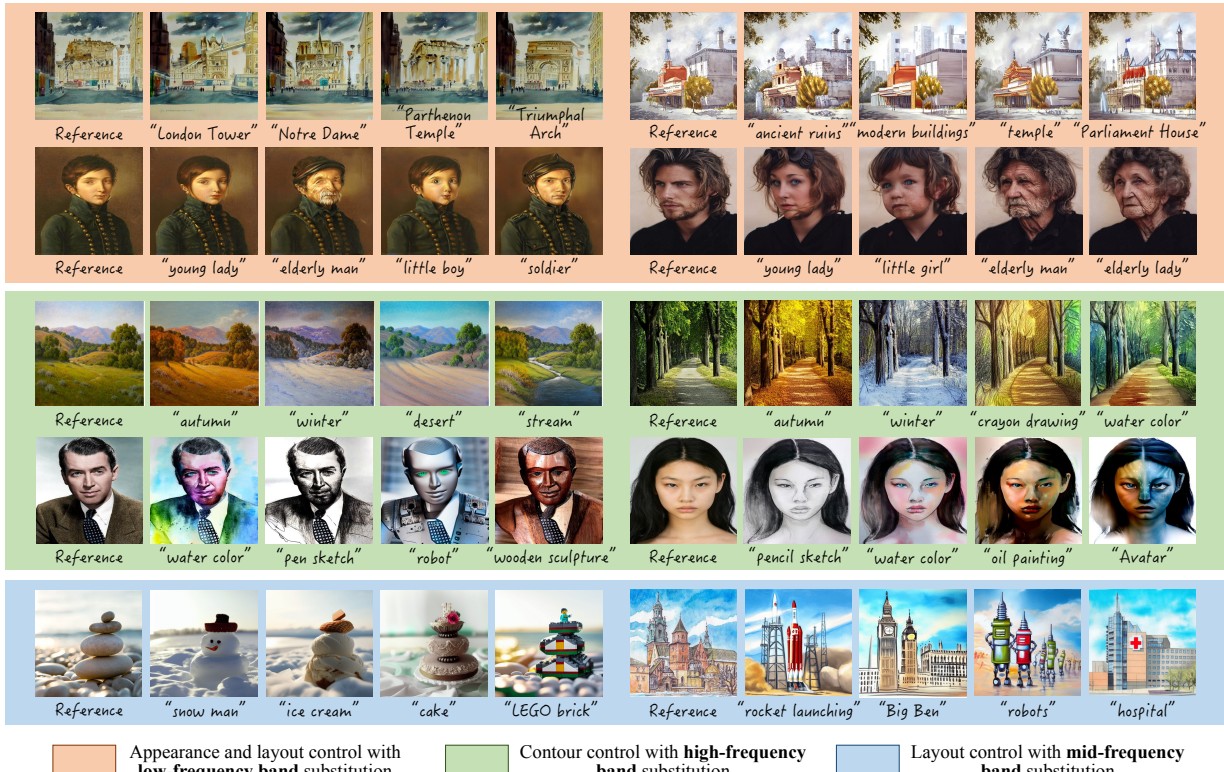

Appearance and layout control with **low-frequency band** substitution

Contour control with **high-frequency band** substitution

Layout control with **mid-frequency band** substitution

Figure 4: Qualitative results of our method with different types of frequency band substitution.

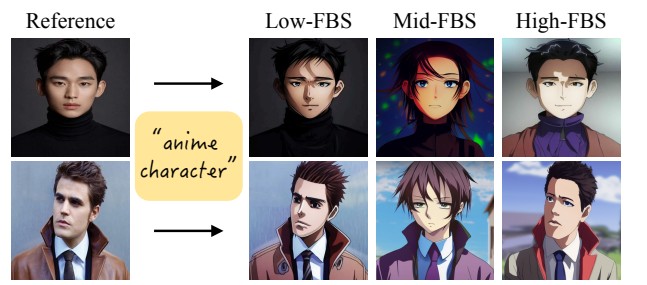

Figure 5: Comparison between the control effects achieved by the low-FBS, mid-FBS, and high-FBS, respectively.

**Text prompt:** "picture of a robot"

Reference — Diversified sampling results of our **FBSDiff**

Reference — Unique sampling result of **Null-text Inversion**

Figure 6: Our method allows diverse sampling results for fixed reference image and target-text prompt.

## 4.1 Qualitative Results

Example text-driven I2I translation results of our method are shown in Fig. 4. Our method effectively decomposes different guiding factors of the reference image by dynamically substituting corresponding types of DCT frequency bands during sampling. The low-FBS transfers low-frequency information of the reference image into the sampling trajectory, making the generated result inherit the original image appearance and layout. In the mode of high-FBS, high-frequency components of the reference image are transplanted, the resulting generated image is aligned with the reference image in high-frequency contours while the low-frequency appearance

is not restricted. The mid-FBS mainly enforces image layout control by filtering out lower-frequency appearance information and higher-frequency contour information of the reference image in the DCT domain. For all three types of frequency band substitution, the image translation results exhibit high visual quality and high fidelity to the text prompts, both for real-world and artistic reference images. The effect of guiding factor decomposition and control is more clearly demonstrated in Fig. 5. Given a target text for image manipulation, FBSDiff preserves image appearance with low-FBS, maintains image contour while allowing appearance change with high-FBS, and constrains only image layout with mid-FBS.

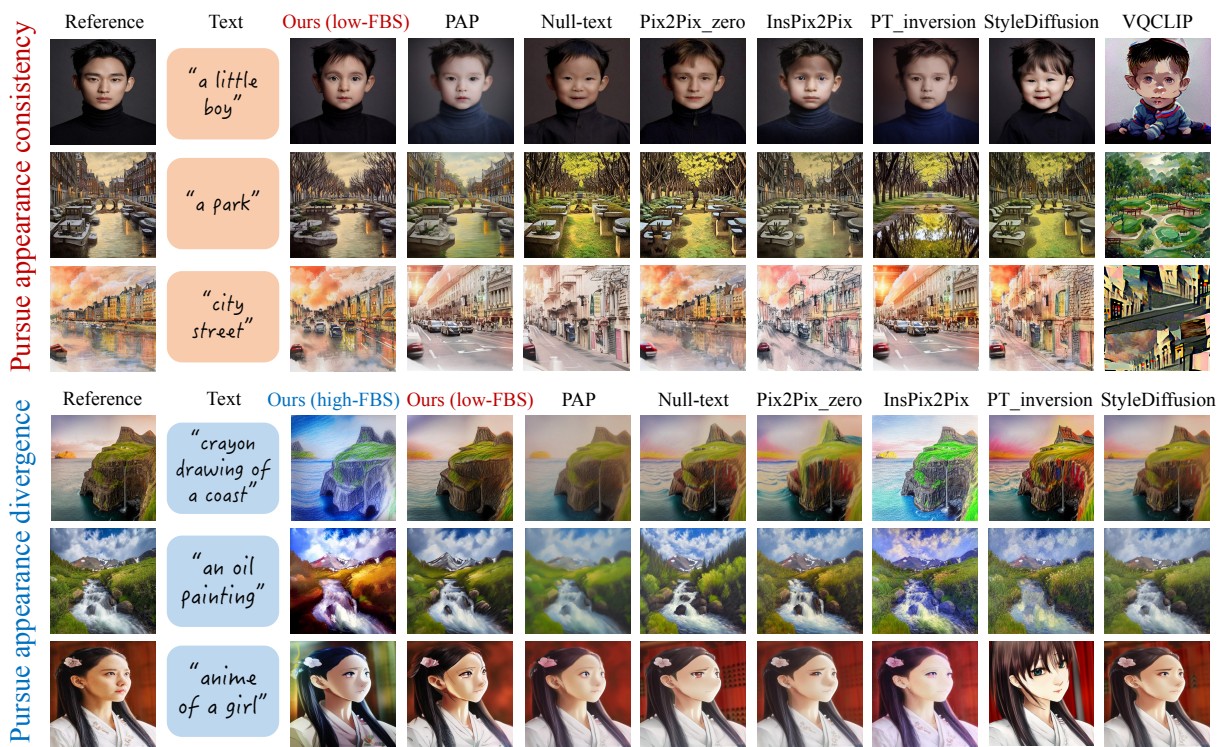

Figure 7: Qualitative comparison of our method with related approaches.

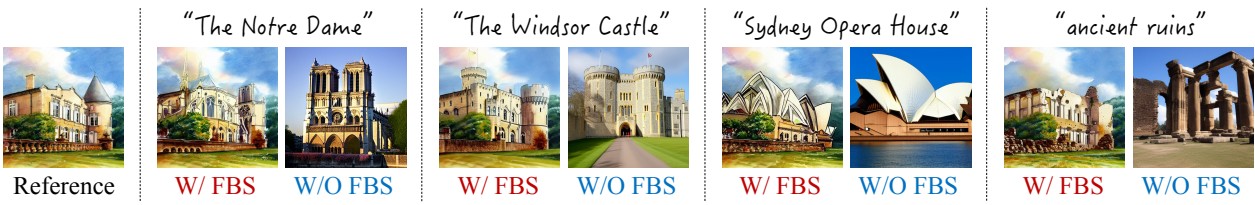

Figure 8: Ablation study results of our method with and without our proposed frequency band substitution.

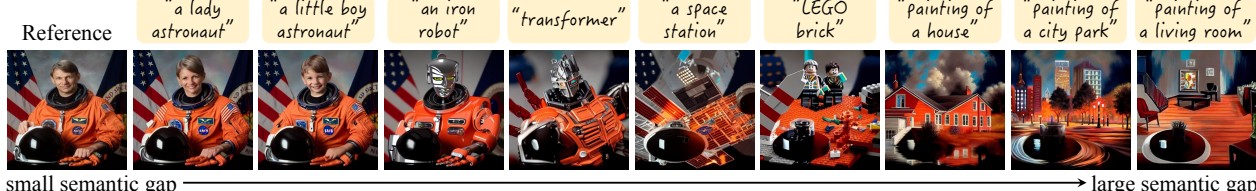

Figure 9: Our method well adapts to the target text with varying semantic discrepancy between the reference image.

In figure 7, we qualitatively compare our method with SOTA text-driven I2I translation methods. Results in the top panel show that our method with low-FBS achieves better appearance consistency between the reference and the translated image than related approaches, and is thus more suitable to image creation scenario where we want to largely borrow the appearance and style from an existing image. Results in the bottom panel show that existing

SOTA inversion-based methods struggle at producing text-driven I2I results that largely deviate from the reference images in visual appearance, while our method with high-FBS enables to generate the translated images with noticeably different appearance, and is thus more suitable to image creation scenario where appearance divergence is pursued. Besides, an advantage of our approach over

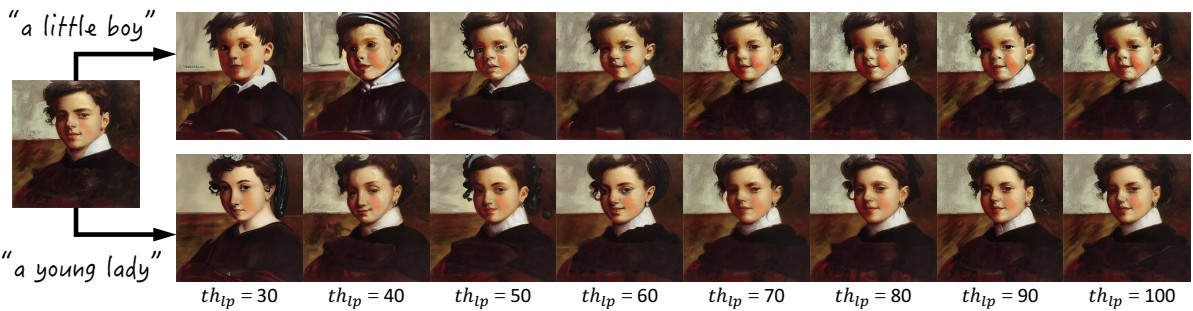

**Figure 10: Demonstration of our method in controlling the low-frequency information guiding intensity by varying the $th_{lp}$ in the low-FBS.**

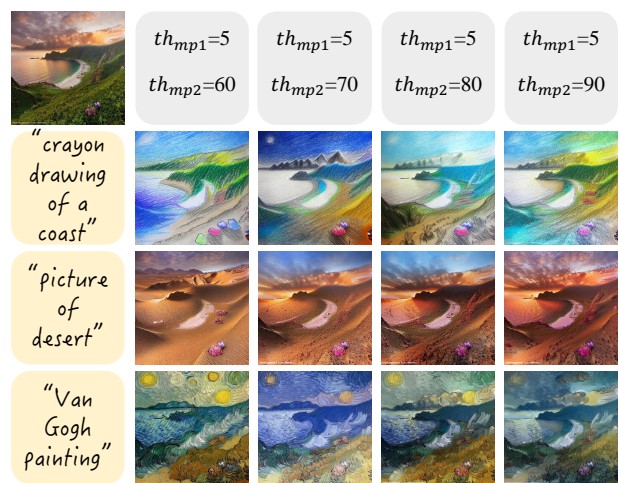

**Figure 11: Demonstration of our method in controlling the high-frequency information guiding intensity by varying the $th_{mp2}$ in the mid-FBS.**

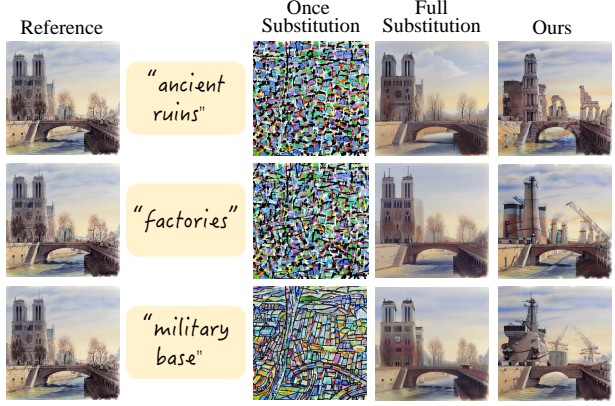

**Figure 12: Ablation study of our method in different FBS manner.**

related methods is sampling diversity. As displayed in Fig. 6, our FB-SDiff can produce diverse I2I results for fixed reference image and target text due to randomly sampling $\tilde{x}_T$ from isotropic Gaussian distribution, while other inversion-based methods [8, 18, 23, 26, 40] lack such sampling diversity for directly initializing $\tilde{x}_T$ with the inverted image embedding. The reference image control functionality of FBS is clearly shown in Fig. 8, in which we see that removing FBS leads to the sampled results with no correlation with the reference image. Moreover, as Fig. 9 displays, our method can well adapt to the text prompts with varying semantic discrepancies with the reference image, producing I2I results that comply with the target text even in the cases of large semantic discrepancy.

Besides the controllability in the guiding factors of the reference image, the guiding intensity is also controllable by modulating the bandwidth of the substituted frequency band. Results displayed in Fig. 10 demonstrate the appearance consistency control of our method by adjusting the low-pass filtering threshold $th_{lp}$ in the low-FBS. In this case, larger value of $th_{lp}$ corresponds to wider bandwidth (more information) of the transplanted frequency band, leading to I2I results with more resemblance to the reference image, while lower value of $th_{lp}$ brings more variations of the generated results to the reference images. Likewise, results in Fig. 11 demonstrate the structure consistency control of our method by tuning the mid-pass filtering upper bound threshold $th_{mp2}$ in the mid-FBS. When increasing the value of $th_{mp2}$, more high-frequency information of the reference image is included and transplanted, leading to more accurate contours of the reference image transferred into the generated result. Decreasing the value of $th_{mp2}$, on the contrary, shrinks the transplanted high-frequency information, and thus leads to weaker structure consistency of the generated images.

## 4.2 Ablation Study

We also explore other designs of frequency band substitution, including substituting the frequency band only once at $\lambda T$ time step rather than along the whole calibration phase (which we denote as **Once Substitution**) and substituting the full DCT spectrum rather other a partial frequency band of it (which we refer to as **Full Substitution**). Results in Fig. 12 show that Once Substitution fails to produce reasonable images, indicating that step-by-step FBS along the whole calibration phase is of crucial importance for smooth information injection and stable information fusion. Since image content is basically formed in the early stage of the diffusion sampling process, removing feature calibration of FBS in the early

**Table 1: Quantitative evaluations of the text-driven I2I translation methods.**

| Emphasis | Pursuing image appearance consistency | | | | | Pursuing image appearance divergence | | | |
|---|---|---|---|---|---|---|---|---|---|
| Metrics
Methods | Structure
Similarity(↑) | LPIPS(↓) | AdaIN Style
Loss(↓) | CLIP
Similarity(↑) | Aesthetic
Score(↑) | Structure
Similarity(↑) | AdaIN Style
Loss(↑) | CLIP
Similarity(↑) | Aesthetic
Score(↑) |
| PAP | 0.954 | 0.278 | 20.525 | 0.316 | 6.583 | 0.957 | 27.848 | 0.306 | 6.439 |
| Null-text | 0.950 | 0.247 | 17.627 | 0.310 | 6.514 | 0.952 | 25.667 | 0.293 | 6.325 |
| Pix2Pix_zero | 0.952 | 0.242 | 16.745 | 0.308 | 6.490 | 0.955 | 25.152 | 0.295 | 6.287 |
| InstructPix2Pix | 0.959 | 0.244 | 25.796 | 0.312 | 6.266 | 0.960 | 29.245 | 0.286 | 6.195 |
| PT_inversion | 0.946 | 0.249 | 22.926 | 0.313 | 6.481 | 0.951 | 26.585 | 0.292 | 6.269 |
| StyleDiffusion | 0.945 | 0.251 | 24.667 | 0.311 | 6.497 | 0.944 | 30.344 | 0.290 | 6.255 |
| **FBSDiff (ours)** | 0.962 | 0.240 | 15.302 | 0.314 | 6.566 | 0.958 | 34.725 | 0.309 | 6.464 |

stage inevitably leads to large mismatch between the sampling and the reconstruction trajectory. This causes remarkably incoherent DCT space after applying FBS at an intermediate time step and thus leads to abnormal results. Besides, it shows that Full Substitution fails to manipulate image semantics as per the text. This is because substituting the full DCT spectrum is equivalent to absolute feature replacement, which makes the sampling trajectory in the early calibration phase totally the same as the reconstruction trajectory. Therefore, the content of the generated image has already been formed to be basically the same as the reference image after the calibration phase, making it difficult to manipulate image semantics in the latter non-calibration phase.

## 4.3 Quantitative Evaluations

For quantitative evaluation, we separately evaluate all the methods on the text-driven I2I translation task pursuing image appearance consistency (favoring appearance preservation) and the task pursuing image appearance divergence (favoring large modification of image appearance). For the former task, we assess model performance by measuring structure similarity (↑), perceptual similarity (↑), and style distance (↓) between each pair of the reference and translated image. For the latter task, we assess models' appearance modification and structure-preserving abilities by measuring structure similarity (↑) and style distance (↑) between the reference and the translated image pairs. Besides, CLIP similarity (↑) is evaluated for both two tasks to measure semantic consistency between the target text and the translated image, i.e., the text fidelity of the I2I translation results. Finally, we evaluate the aesthetic score of the translated images with the pre-trained LAION Aesthetics Predictor V2 model. We use the DINO-ViT self-similarity distance [39] as the metric for structure similarity, use LPIPS [44] metric for perceptual similarity, and use AdaIN style loss [14] to measure style discrepancy between the reference and translated image.

We perform all the quantitative evaluations on the LAION Aesthetics 6.5+ dataset, we sample 300 evaluation images for each task and manually design 2 editing texts for each image, resulting in 600 evaluation samples per task. The average values of all the metrics are shown in Tab. 1. For evaluation of our method, we use the low-FBS for the I2I task pursuing appearance consistency, and use the high-FBS for the task pursuing appearance divergence. Results shown in the table show that our method achieves top rankings for all the metrics, demonstrating the superiority of our method over related approaches. Among all the compared methods, our

method is the only one allowing versatile control over the guiding factors of the reference image to the generated image, and also takes advantage in continuous control over the guiding intensity of the reference image.

## 5 CONCLUSION

This paper proposes a novel plug-and-play module adapting the pre-trained text-to-image diffusion model to versatile I2I applications. At the heart of our method is a training-free frequency band substitution layer, which dynamically calibrates the denoising diffusion process of the target image by substituting certain DCT frequency band extracted from the reference image into the reverse sampling diffusion process. We demonstrate that versatile I2I applications can be unified by our method simply by switching among different modes and ranges of the substituted frequency band, realizing effective, flexible, and comprehensive control over the translated images.

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
