# OpenReview forum: "FBSDiff: Plug-and-Play Frequency Band Substitution of Diffusion Features for Highly Controllable Text-Driven Image Translation"
_acmmm.org/ACMMM/2024/Conference — MM2024 Poster_

### Official Review · Reviewer_adM2 · 2024-05-19

**Rating:** 4
**Confidence:** 3

**Summary:**

This  paper proposes a simple training-free method to advance controllability in Text-to-Image (T2I) diffusion models, by substituting frequency bands in the DCT spectral space, enabling the dynamic replacement of frequency features with those of a reference image. Specifically, the reference image undergoes inverse diffusion and diffusion processes to obtain the frequency features for substitution. Experimental comparisons demonstrate superior performance in image translation quality and versatility.

**Strengths:**

1. The proposed frequency band substitution framework is novel for text-conditioned I2I translation.
2. The authors provide extensive qualitative and quantitative experiment results to support the effectiveness of their method.
3. The writing is clear and easy-to-follow.

**Limitations:**

1. Could the authors provide analysis and empirical results of the capability limit of modifying frequency bands? While frequency domain is useful for handling structural level information, it may be inferior in terms of semantic information, which seems to me the major limitation of this work,.
2. The inversion and reconstruction processes of the reference image seem to introduce extra computation burden. Could the authors also provide comparison of training and inference costs with other methods?
3. It would be more convincing to see more quantitative ablation results on the hyperparameters to define low, mid, and high level frequency bands. Is the performance of FBSDiff robust to different frequency band settings?

**Suitability:**

3

---

### Official Review · Reviewer_rth4 · 2024-05-23

**Rating:** 4
**Confidence:** 2

**Summary:**

This paper introduces a novel approach that utilizes various frequency bands of diffusion features in the DCT spectral space, enabling dynamic frequency band substitution to guide image generation. This technique realizes highly controllable and versatile text-driven image-to-image (I2I) translation.

**Strengths:**

1. The introduction of the Non-Calibration phase and FBS based on the PAP method further enhances controllability.
2. The proposal to use different frequency bands to distinguish different types of features is innovative.
3. Both quantitative and qualitative comparisons demonstrate impressive results.

**Limitations:**

1. The quantitative comparison lacks the FID metric, which is commonly used to measure image fidelity in text-to-image methods.
2. The introduction of the Non-Calibration phase adds an additional tunable hyperparameter \(\lambda\) compared to the PAP method, which seems to reduce ease of use during generation.
3. Line 430 claims that the low-frequency band can control color, but the experiments in Figure 5 show that the color did not change in the Low-FBS.

**Suitability:**

3

---

### Official Review · Reviewer_JM2e · 2024-05-24

**Rating:** 3
**Confidence:** 4

**Summary:**

This work proposes a plug-and-play and controllable text-driven image2image translation method, FBSDiff, from a frequency-domain perspective. FBSDiff first encodes the reference image and computes the inversion trajectory. Then, it computes the latent maps for both the reconstruction trajectory and the sampling trajectory. By leveraging the frequency band substitution layer, FBSDiff calibrates sampling latent maps with different frequency band extractions corresponding to different controls of appearance, layout, and contour. Comprehensive experiments show the effectiveness of the method.

**Strengths:**

- The method is training or finetuning free.
- Comprehensive experiments have been done.
- FBSDiff can do large semantic gap translation.

**Limitations:**

- Though the method is training free, it requires computing multiple trajectories. The computing cost is not really low.
- Authors claim different frequency bands correspond to different controls, i.e. low frequency to appearance and layout, high frequency to contour, and mid frequency to layout. The correspondence is not very clear. More results should be shown to make it convincing.
- Due to the aforementioned shortages, it seems several trials on each image should be conducted to achieve the best translation result, which limits the application of this method.
- The performance of FBSDiss is not obviously superior to other methods.

**Suitability:**

3

---

### Meta-Review · Senior_Area_Chairs · 2024-07-09

**Recommendation:** Accept (Poster)
**Confidence:** 5

**Metareview:**

This paper proposes a method for high-quality and versatile text-driven image translation which at generation time is done without any model training, model fine-tuning, or online optimization. Reviewers indicate that there is a comprehensive set of experiments and that both quantitative and qualitative results are very good. There were some initial doubts, but after the rebuttal all reviewers indicate this as a borderline or weak accept.